# The *Aspergillus fumigatus* Extracellular Polysaccharide Galactosaminogalactan Displays Context-Dependent Cooperative and Competitive Social Traits in Mixed Biofilms

**DOI:** 10.3390/jof11100695

**Published:** 2025-09-25

**Authors:** Longyun Cong, Yufei Zhang, Hua Chen, Ruiyang Lu, Shizhu Zhang

**Affiliations:** Jiangsu Key Laboratory for Pathogens and Ecosystems, Jiangsu Engineering and Technology Research Center for Microbiology, College of Life Sciences, Nanjing Normal University, Nanjing 210023, China

**Keywords:** *Aspergillus fumigatus*, biofilms, galactosaminogalactan, social characteristics

## Abstract

Biofilm-dwelling cells construct communities by secreting extracellular polysaccharide (EPS). In bacteria, EPS can act as cooperative public goods or competitive traits, yet the social nature of EPS in fungi remains poorly understood. Galactosaminogalactan (GAG) is an EPS produced by the human-pathogenic fungus *Aspergillus fumigatus*. The study of social characteristics of GAG revealed that under basal conditions, GAG can be shared between GAG production strain (GAG^+^) and non-production strain (GAG^−^) in mixed biofilms. This led to significant competitive advantages for GAG^−^, with fitness outcomes dependent on initial inoculum ratios, cultivation duration, and nutrient availability. Conversely, under cell wall stress induced by antifungal drug caspofungin, GAG confers a competitive advantage for GAG^+^ in the mixed biofilms. Further investigation revealed that GAG^+^ cells are able to retain higher levels of GAG on the hyphal surface compared to GAG^−^ in the mixed biofilms. This hyphal surface-associated GAG layer might protect GAG^+^ from caspofungin-mediated damage, creating a lineage-specific competitive advantage. Overall, GAG has a dual-trait social nature in biofilms, functioning as a public good at the population level and as a competitive trait for the producing lineage, switching according to environmental conditions.

## 1. Introduction

Biofilms are structured microbial communities encased in a self-generated extracellular matrix composed of proteins, extracellular DNA, lipids, and extracellular polysaccharides (EPS) [1]. The EPS can provide many diverse benefits to the cells in the biofilms, including adhesion, protection, and structure [2,3]. Importantly, these secreted factors have social characteristics, acting as cooperative public goods or competitive traits [4,5].

Biofilms are believed to necessitate substantial cooperation, exemplified by the fact that EPS serve as a communal resource generated by one cell, which other cells can utilize [6]. While cooperation enhances group dynamics, it also faces risks from opportunistic cheaters [4,5,7]. Findings from bacterial systems indicate various approaches to prevent the exploitation of bacteria that produce public goods. These approaches include restricting cooperation to genetically related individuals, often through mechanisms like kin discrimination or population bottlenecks [8,9]. Another strategy involves the ongoing diversification of shared compounds to deter exploitation by different alleles [10,11]. Additionally, structuring the population spatially increases the likelihood that producers will be surrounded by fellow cooperators [12,13]. Moreover, preventing cheating may arise from genetic mutations; for instance, a specific mutation in *rapP* instigates cheating deterrence in *Bacillus subtilis* by reducing the overall production of the public good surfactin [14].

Alternatively, certain secreted factors can confer a competitive edge, selectively benefiting their genetically identical clonemates. In *Vibrio cholerae* biofilms formed in microfluidic devices, extracellular polymeric substances-producing cells offer selective benefits to their clonemates and gaining a significant edge in competition against an isogenic strain lacking EPS [15]. Distinct from above finding, the *Pseudomonas aeruginosa* PSL (an exopolysaccharide encoded by the polysaccharide synthesis locus) is social and is shared with other nonproducing cells, but PSL is non-cheatable in mixed biofilms [7]. Collectively, the research on bacteria indicates that there is considerable variability in the social dynamics of different types of EPS across various species [16]. The production of EPS is not limited to bacteria, as fungi are also capable of producing it. However, in contrast to the research in bacterial systems, the social characteristics of EPS in fungi remain poorly understood.

*Aspergillus fumigatus* is a filamentous fungal pathogen that causes life-threatening invasive infections in immunocompromised hosts. The biofilms formed by *A*. *fumigatus* exhibit distinctive morphological and architectural features that set them apart from those associated with bacterial and yeast biofilms [17,18]. Biofilm-mediated resistance involves multiple mechanisms, including efflux pump overexpression and metabolic adaptation to hypoxia [18]. Analyses involving three-dimensional surface plotting have demonstrated that *A. fumigatus* biofilms display spatially organized hyphae, well-defined hyphal channels, and vertical growth patterns of hyphae as their distinguishing traits [19,20]. Within the extracellular matrix of *A. fumigatus* biofilms, galactosaminogalactan (GAG), a specific glycan, is produced both in laboratory settings and within living organisms [21,22]. The production of GAG is regulated by a gene cluster comprising five genes located on chromosome 3 [23,24]. This GAG biosynthetic gene cluster has been identified in numerous fungal species including several plant and human fungal pathogens. Notably, it is absent in *Saccharomyces cerevisiae* and the human fungal pathogen *Candida albicans* [25]. GAG is found not only on the *A. fumigatus* hyphal surfaces but is also released as an element of the extracellular matrix. Consequently, GAG fulfills various roles in the interactions between *A. fumigatus* and the immune system of the host and is regarded as a significant factor contributing to fungal virulence [22,26,27]. Being an essential element of the extracellular matrix, the development of *A. fumigatus* biofilms relies on the synthesis of GAG, and the loss of GAG prevents adherent biofilms formation [24,28]. Recently, research has explored the role of GAG in mixed biofilms formed by *P*. *aeruginosa* and *A. fumigatus*, revealing that GAG plays a crucial role in mediating bacterial adhesion to fungal hyphae [29]. Nevertheless, the social characteristics of GAG within biofilms communities remain unclear.

Here, the social nature of GAG was investigated in *A. fumigatus* biofilms. We discovered that GAG displays a distinctive social property with dual traits. Specifically, depending on different environmental conditions, it can act as a public good as well as a competitive trait.

## 2. Materials and Methods

### 2.1. Strains and Growth Culture Conditions

Strains used in this study are listed in Appendix A. *Aspergillus fumigatus* AF1160 (Δ*ku80*, *pyrG*) was purchased from the Fungal Genetics Stock Center; its complemented strain, AF1161 (AF1160::*pyrG*) [30], was used as the parental wild type (WT). All *A. fumigatus* strains were cultured on solid yeast extract, agar and glucose (YAG) medium [glucose, 20 g L^−1^; yeast extract, 5 g L^−1^; 1 mL L^−1^ trace elements (ZnSO_4_·7H_2_O, 2.2 g L^−1^; H_3_BO_3_, 1.1 g L^−1^; MnCl_2_·4H_2_O, 0.5 g L^−1^; FeSO_4_·7H_2_O, 0.5 g L^−1^; CoCl_2_·5H_2_O, 0.16 g L^−1^; CuSO_4_·5H_2_O, 0.16 g L^−1^; (NH_4_)_6_Mo_7_O_24_·4H_2_O, 0.11 g L^−1^; Na_4_EDTA, 5 g L^−1^)] to produce conidia [30,31]. In general, the biofilms of *A. fumigatus* were cultured on minimal medium (MM) [50 mL L^−1^ Nitrate Salts (NaNO_3_, 120 g L^−1^; KCl, 10.4 g L^−1^; MgSO_4_·7H_2_O, 10.4 g L^−1^; KH_2_PO_4_, 30.4 g L^−1^), 1 mL L^−1^ trace elements and glucose, 10 g L^−1^, pH 6.5] [30]. For glucose starvation conditions, prepared MM containing 0.05% glucose (0.5 g L^−1^); for iron starvation conditions, MM was prepared with iron-depleted trace elements (excluding FeSO_4_·7H_2_O).

### 2.2. Strains Construction

Gene *uge3* encodes a key isomerase in the biosynthesis of GAG. To construct the deletion strain of *uge3*, fusion PCR was used to construct the *uge3* knockout cassette as previously described [32]. In brief, approximately 1 kb sections of regions flanking the *uge3* gene were amplified with the primers Δ*uge3*-P1/P3 and Δ*uge3*-P4/P6, respectively. The hygromycin B resistance gene *hph* from the plasmid pAN7-1 was amplified with the primers *hph*-F/R. Subsequently, these three PCR products served as templates to generate the complete *uge3* deletion cassette via another round of PCR with the primers Δ*uge3*-P2/P5, and then transformed into the parental *A. fumigatus* strain AF1161. Transformants were verified by diagnostic PCR with the primers Δ*uge3*-SF/SR, Δ*uge3*-P1/*hph*-down, and *hph*-up/Δ*uge3*-P6, respectively.

To construct the complemented strains *uge3*^C^ and *uge3*^C-RFP^, approximately 2.6 kb sections of the *uge3* gene promoter and ORF were amplified with the primers *uge3*^C^-P1/P3, and approximately 1 kb sections of the *uge3* gene terminator were amplified with the primers *uge3*^C^-P4/P6. The phleomycin resistance gene *phel* was amplified with the primers *phel*-F/R. Subsequently, these three PCR products served as templates to generate the complete *uge3* complement cassette via another round of PCR with the primers *uge3*^C^-P2/P5, and then transformed into the strains Δ*uge3* and Δ*uge3*^RFP^, respectively. Transformants were verified by diagnostic PCR with the primers *uge3*^C^-P1/*phel*-down, and *phel*-up/*uge3*^C^-P6, respectively.

To generate the reporter strain WT^GFP^, the GFP was fusion expressed with glyceraldehyde-3-phosphate dehydrogenase (GAPDH). Briefly, a *gfp*+*pyrG* fragment was amplified from plasmid pFNO3 with the primers GFP+*pyrG*-F/R. Approximately 1 kb fragments upstream and downstream of the *gapdh* stop codon were amplified with the primers GAPDH-P1/P3 and GAPDH-P4/P6, respectively. These fragments were fused by PCR with the primers GAPDH-P2/P5, and the PCR product was used to transform strain AF1160. Homologous integration was verified by PCR with the primers GAPDH-P2/GFP+*pyrG*-down and GFP+*pyrG*-up/GAPDH-P5, respectively. To generate the reporter strain Δ*uge3*^RFP^, the strain WT^RFP^ was firstly constructed by using a similar strategy. And then, the *uge3* knockout cassette was transformed into WT^RFP^ to generate the strain Δ*uge3*^RFP^. All of the primers that were used in this study are listed in Appendix A. The construction processes of all strains used in this study are shown in Appendix A.

### 2.3. Confocal Laser Scanning Microscopy of Fungal Biofilms

*A. fumigatus* biofilms were cultured for imaging in glass bottom cell culture dishes (Φ20 mm, NEST, Wuxi, China). For the mixed biofilms, the indicated conidia were mixed at indicated ratios to achieve a total conidial density of 2 × 10^5^ conidia per mL of MM. The mixture was statically co-cultured in glass bottom cell culture dishes (0.5 mL per dish) at 37 °C. The exact number of indicated conidia was determined by plate counting. Fluorescence confocal microscopy was performed on a Nikon, A1R with CFI Plan Apochromat 10× C Glyc. The biofilms were imaged over time with z-stack intervals of 1.75 μm. Images were acquired and analyzed using NIS viewer 5.22 and ImageJ 1.54 p software.

### 2.4. Crystal Violet Assay for Biofilm Biomass Determination

The visualization and quantification of *A. fumigatus* biofilms were carried out in accordance with previously established methods [33]. In brief, the indicated conidia were mixed at indicated ratios to achieve a total conidial density of 1 × 10^5^ conidia per mL of MM. The mixture was statically co-cultured in 24-well microplates (0.5 mL per well) at 37 °C. The 24-well microplates were tilted, and the upper-layer medium was gently discarded. The biofilms were rinsed twice with 1 mL of distilled water per well. Then, 0.25 mL of 0.1% crystal violet staining solution was added to each well and incubated for 10 min to allow staining.

Following this, the crystal violet staining solution was discarded, and the residual staining solution was rinsed twice with 1 mL of distilled water per well. After draining off the excess distilled water, 0.5 mL of absolute ethanol was added to each well for decolorization for 10 min. Finally, the decolorization solution in the wells was mixed gently, and 75 μL of the decolorization solution per well was placed into a 96-well microplate. The optical density of the decolorization solution was assessed at 600 nm.

### 2.5. Quantification of Ratios of GAG^+^ and GAG^−^ in Mixed Biofilms by Quantitative PCR (Q-PCR)

To quantify the ratios of GAG^+^ and GAG^−^ in mixed biofilms, the indicated conidia were mixed at indicated ratios to achieve a total conidial density of 1 × 10^5^ conidia per mL of MM with 25 mL per dish (Φ90 mm). Biofilms were co-cultured under the indicated experimental conditions, and genomic DNA was extracted from the harvested biofilms (Fungal genome rapid extraction kit, B518229, Sangon Biotech, Shanghai, China). WT^GFP^ and Δ*uge3*^RFP^ DNA was specifically quantified by Q-PCR with primers for *gfp* and *rfp*, respectively. The relative proportion of Δ*uge3*^RFP^ in the mixed biofilms was calculated by formulas (*N*_0_ = the initial concentration of genes in the genomic DNA of mixed biofilms):
The relative proportion of Δuge 3RFP=N0,rfpN0,rfp+N0,gfp.

The 20 μL Q-PCR mixture was composed of 10 μL 2× ChamQ SYBR qPCR Master Mix (Low ROX Premixed) (Q331-02, Nanjing Vazyme, Nanjing, China), 0.8 μL of each primer (10 μM), and 5 μL of template DNA (5 ng μL^−1^). Cycling was performed on the ABI 9600 fast real-time PCR system with an initial hold at 95 °C for 30 s, followed by 40 cycles at 95 °C for 10 s and 60 °C for 30 s, with a cycle threshold of 35. Negative controls without DNA were included in each Q-PCR run. All primers used for Q-PCR in this study are listed in Appendix A.

### 2.6. XTT Assay

For *A. fumigatus* mixed biofilms, the indicated conidia were mixed at indicated ratios to achieve a total conidial density of 1 × 10^5^ conidia per mL of MM. The mixture was statically co-cultured in 24-well microplates (0.5 mL per well) at 37 °C for 24 h. After 24 h of culture, biofilms were rinsed once with 0.5 mL of 1× PBS. Biofilms were then treated with indicated concentrations of voriconazole (S31125, Shanghai yuanye, Shanghai, China), amphotericin B (V900919, Sigma, St. Louis, USA), caspofungin (S26841, Shanghai yuanye), micafungin (S89567, Shanghai yuanye) and calcofluor white (S22603, Shanghai yuanye) for 12 h. After 12 h of treatment, the MM with drugs was removed and the biofilms were rinsed once again with 0.5 mL of 1× PBS. XTT [2,3-bis-(2-methoxy-4-nitro-5-sulfophenyl)-2H-tetrazolium-5-carboxanilide] solution (100 μg XTT sodium salt per mL of 1× PBS with 6.25 μM menadione) (XTT sodium salt, S30625, Shanghai yuanye; menadione, A502486, Sangon Biotech) was added at 0.625 mL per well and statically incubated at 37 °C for 2 h in a dark chamber. The supernatant of the XTT solution was placed into a 96-well microplate, and the optical density was assessed at 460 nm. To determine the metabolic activity inhibition, drug-treated wells were compared against untreated ones.

### 2.7. Microscopy

#### 2.7.1. Observation of Hyphal Morphology Under Treated with Caspofungin

In total, 5 × 10^4^ conidia of WT and Δ*uge3* mutant were separately cultured in cover glass with 0.5 mL MM at 37 °C for 10 h, then treated with caspofungin for an additional 6 or 24 h. Since the Δ*uge3* mutant could not adhere to the growth medium, 0.5 mL of collagen coating solution (Sigma) was applied to cover glass overnight at 4 °C. The solution was removed prior to inoculation. The strain was then rinsed twice with 1× PBS and produced as squash slides. Images were obtained using a Zeiss Axiom Imager A1 microscope (Zeiss, Jena, Germany).

#### 2.7.2. GAG Polysaccharide Characterization

In this study, a specific fluorescein-tagged soybean agglutinin lectin, designated as SBA-FITC, was employed as a crucial tool for the analysis of GAG polysaccharide production [28]. Briefly, the indicated conidia were mixed at a 1:1 ratio to obtain a total conidia density 1 × 10^5^ conidia per mL, which were co-cultured on cover glass in MM at 37 °C for 12 h. The hyphae samples were rinsed twice with 0.5 mL of D-PBS to ensure the removal of any residual substances. After rinsing, the hyphae were stained with SBA-FITC (Vector Labs, Burlingame, CA, USA) under light-protected conditions. Following a 3 h incubation on ice, the hyphae samples were rinsed twice again with 0.5 mL of D-PBS. Finally, images of the stained samples were obtained using a Zeiss Axiom Imager A1 microscope.

## 3. Results

### 3.1. GAG Provides Social Benefits in Mixed Biofilms

Uge3 is a key isomerase in GAG biosynthesis. Previous studies showed that losing *uge3* causes GAG production defects and prevents adherent biofilms formation [24]. To observe if GAG-producing strains benefit non-producing strains in mixed biofilms, wild-type *A. fumigatus* with a GFP (green fluorescent protein)-expressing construct (WT^GFP^, GAG^+^) and an isogenic Δ*uge3* mutant with an RFP (red fluorescent protein)-expressing construct (Δ*uge3*^RFP^, GAG^−^) were generated. Consistence with previous reports, confocal laser scanning microscope (CLSM) images showed that *A. fumigatus* WT^GFP^ biofilms are mainly composed of branched hyphae with a dense mat of filaments at the base perpendicular to the vertical axis. Above approximately 50 μm, the filaments grow polarized towards the air-liquid interface with little deviation from the vertical axis up to 1 mm (Figure 1A). In comparison, the biomass distribution in the Δ*uge3*^RFP^ is altered such that most interconnected vegetative hyphae grow in the form of floating with varying degrees of deviation from the vertical axis. Relatively sparse filaments from the base up to the half of the total volume (500 μm) were observed (Figure 1A). Strikingly, when co-culture GAG^+^ and GAG^−^, hyphae of GAG^−^ (red) can co-aggregate with hyphae of GAG^+^ (green), forming stable WT-like mixed biofilms at varied initial ratios (the initial proportion of GAG^−^ is 10%, 50% and 90%, respectively) (Figure 1A).

The crystal violet assay showed that GAG^−^ proportion negatively affected the total biomass. When the initial proportion of GAG^−^ exceeded 60%, the total biomass of the mixed biofilms decreased (Figure 1B). However, it is important to note that even at a 90% initial proportion of the GAG^−^, the biomass of the mixed biofilms was still significantly greater than that of the Δ*uge3* mutant cultured alone (Figure 1B). Collectively, these results suggest that GAG is shared between GAG^+^ and GAG^−^ in mixed biofilms, with GAG^+^ conferring benefits to GAG^−^.

### 3.2. GAG Production Is an Exploitable Cooperative Behavior in Mixed Biofilms

Next, we investigated whether GAG^−^ could act as “social cheats” in GAG^+^/GAG^−^ mixed biofilms. To this end, we co-cultured GAG^+^ (WT^GFP^) and GAG^−^ (Δ*uge3*^RFP^) at different ratios (the initial proportion of GAG^−^ ranging from 10% to 90%), and the relative fitness was assayed via quantitative PCR. The results showed that GAG^−^ can gain a fitness advantage without contributing to GAG production. Strikingly, the relative fitness of the GAG^−^ displayed significant advantage compared to GAG^+^ in 24 h mixed biofilms with initial proportion of GAG^−^ ranging from 20% to 80%. However, the trend of fitness advantage of GAG^−^ flattens out when the initial proportion of GAG^−^ was above 80% or below 20% (Figure 2A).

Long-term competition (up to 60 h) between GAG^+^ and GAG^−^ in mixed biofilms showed that the fitness advantage of GAG^−^ gradually decreased over time (Figure 2B). The fitness advantage of GAG^−^ mostly disappeared after 60 h when co-cultured with GAG^+^. Nutrient limitation generally affects cooperative behavior by increasing costs [34]. When cultured under glucose-starved (minimal medium containing 0.05% glucose) or iron-limited condition (minimal medium without addition of iron), GAG^−^ had a fitness advantage in long-term competition but not in short-term competition (Figure 2C,D). Thus, the GAG^+^ do not outcompete the GAG^−^ at any initial proportions under all tested conditions. To confirm that the aforementioned results were attributed solely to the loss of GAG production, we constructed the Δ*uge3*^RFP^ complemented strain *uge3*^C-RFP^ (Appendix A). The results showed that *uge3*^C-RFP^ displayed nearly identical fitness to WT^GFP^ across all tested initial proportions and nutrient conditions (Appendix A).

Overall, these results indicate that GAG is a public good shared in mixed biofilms. GAG^−^ can outcompete GAG^+^, and GAG^−^ behavior pattern matched the core characteristics of “social cheats” in mixed biofilms, but the extent of their competitive advantage depends on the initial inoculum ratios, cultivation duration, and nutrient availability.

### 3.3. GAG Confers a Competitive Trait Under Caspofungin Treatment in the Mixed Biofilms

We further tested the relative fitness of GAG^+^/GAG^−^ in mixed biofilms under antifungal drugs treatment. Considering the strong inhibitory effects of those antifungal drugs on the germination, we firstly co-cultured equal amounts of GAG^+^/GAG^−^ for 12 h on minimal medium to form young mixed biofilms, then treated them with antifungal drugs for another 24 h (Figure 3A). Given drug treatment inhibits the growth of mixed biofilms to varying degrees (Figure 3B), we hypothesized that the proportion of GAG^−^ after antifungal drugs treatment would be between that of the untreated groups at 12 h and 36 h biofilms. When treated with voriconazole and amphotericin B, the proportion of GAG^−^ was as expected. However, after caspofungin treatment, the relative proportion of GAG^+^ in the mixed biofilms increased significantly (Figure 3C). CLSM confirmed this result; after caspofungin treatment, only a few GAG^−^ cells were visible at the bottom of the mixed biofilms (Figure 3D). This result indicates that GAG^+^ cells receive more protection than GAG^−^ under caspofungin treatment in mixed biofilms.

We further tested whether long-term cultivation would further enhance the fitness advantage of GAG^+^ in the mixed biofilms. However, treating the mixed biofilms with caspofungin for 24, 36, 48, and 60 h did not lead to significant changes in the relative proportion of GAG^+^ (Figure 3E). This observation suggests that GAG^−^ may be entrapped within the matrix during biofilm formation, allowing a small subset of GAG^−^ cells to persist.

### 3.4. GAG Plays Protective Roles in Conferring Tolerance to Cell Wall Stresses

As mentioned in the previous findings, GAG^+^ receive more protection than GAG^−^ under caspofungin treatment in mixed biofilms, we further explored the underlying mechanism. First, we assayed the drug susceptibility of mixed biofilms with GAG^+^/GAG^−^ at varied initial ratios (the initial proportion of GAG^−^ is 10%, 50% and 90%, respectively). Strikingly, the tolerance of the mixed biofilms to caspofungin was inversely proportional to the proportion of GAG^−^ (Figure 4A). However, this phenomenon was not observed with other antifungal drugs such as voriconazole and amphotericin B. The mixed biofilms displayed similar susceptibility to voriconazole and amphotericin B regardless of the initial GAG^+^/GAG^−^ ratios (Appendix A).

Given caspofungin exerts its effect by inhibiting the activity of β-(1,3)-D-glucan synthase in the fungal cell wall [35], we further tested the tolerance of the mixed biofilms with varied initial GAG^+^/GAG^−^ ratios to other cell wall stress reagents. The results demonstrated that the tolerance of mixed biofilms to micafungin and calcofluor white treatment was also inversely proportional to the proportion of GAG^−^ (Figure 4B,C). This finding exhibited the same trend as that observed with caspofungin treatment, indicating a consistent pattern in the response of mixed biofilms to these cell wall stress agents. This initial ratios depended susceptibility phenotype to caspofungin, micafungin and calcofluor white was not observed in WT and *uge3*^C^ mixed biofilms (Appendix A). Thus, those results provide strong evidence that the amount of GAG is highly significant in relation to the tolerance to cell wall stress agents. To further assess whether the presence or absence of GAG would affect the sensitivity of *A. fumigatus* to caspofungin under submerged biofilm-forming conditions, we observed the hyphal morphology of WT, Δ*uge3* and *uge3*^C^ under caspofungin treatment. The results showed that caspofungin had a significant inhibitory effect on the hyphae growth of all tested strains, but exhibited a stronger inhibitory effect on Δ*uge3* (Figure 4D). Following 6 h caspofungin exposure, most WT and *uge3*^C^ hyphae maintained structural integrity despite growth arrest, whereas the majority of Δ*uge3* hyphae exhibited early tip lysis. This divergence became more pronounced after 24 h treatment, with near-complete hyphal disintegration in the Δ*uge3* population contrasted by relatively lower amount of tip lysis observed in WT and *uge3*^C^ hyphae (Figure 4D and Appendix A). In contrast, there was no significant difference in caspofungin susceptibility between the Δ*uge3* mutant and WT when measured by the radial growth of dormant conidia (which lack of GAG) on agar plates (Appendix A). These findings suggest that GAG confers protection against caspofungin-induced hyphal damage, and that GAG deficiency renders *A. fumigatus* hyphae more vulnerable to caspofungin.

Considering that GAG^+^ receive more protection than GAG^−^ under caspofungin treatment in mixed biofilms, we speculate that GAG^+^ might retain more GAG, which helps them resist caspofungin. Indeed, a significantly greater amount of GAG was detected on the surface of hyphae of GAG^+^ than that on the hyphal surface of GAG^−^ with or without caspofungin treatment (Figure 4E,F). In some GAG^−^ hyphae, GAG was not detected at locations far from the GAG^+^ regions. Collectively, the results revealed that GAG^+^ are able to retain more GAG on the hyphal surface compared to GAG^−^ in the mixed biofilms, thereby protecting them from cell wall stress-mediated damage.

## 4. Discussion

This study, for the first time, uncovered the social nature of the EPS in the filamentous fungal biofilms. Under basal conditions, GAG is shared between GAG-producing (GAG^+^) and non-producing (GAG^−^) strains, enabling GAG^−^ to co-aggregate with GAG^+^ and form stable biofilms. This cooperation enhances overall community fitness by promoting biofilm structure, a critical advantage in resource-limited or niche-competitive environments. Conversely, under cell wall stress (e.g., caspofungin treatment), GAG^+^ retains more surface-associated GAG, conferring lineage-specific protection and shifting GAG to a competitive trait. This plasticity allows *A. fumigatus* to balance cooperation (for colonization) and competition (for survival under stress), optimizing its persistence in dynamic microbial ecosystems. A working model showing the social nature of GAG in the mixed biofilms is depicted in Figure 5.

Our findings exhibit both similarities and distinctions when compared to the two proposed hypotheses that aim to elucidate the evolutionary stability of EPS. Aligned with the public good hypothesis, our research indicates that GAG production offers a social benefit to GAG^−^, which can effectively outcompete GAG^+^ within mixed biofilms. However, although nonproducing strain can invade wild-type biofilms, the range of the fitness advantage of GAG^−^ remains within a certain limit, and the advantage of GAG^−^ does not expand unrestrictedly. Interestingly, we found that when the initial ratio of GAG^−^ to GAG^+^ is 1:1, the fitness advantage of GAG^−^ is the greatest. Both excessively high and an excessively low initial proportion of GAG^−^ will reduce the fitness advantage of GAG^−^. This mechanism is important because it can prevent the collapse of the mixed biofilms caused by the excessive invasion of GAG^−^.

It has been suggested that limitations in nutrients typically influence cooperative behavior by raising its expenses, as resources need to be reallocated from growth to cooperative actions [34,36,37]. Carbon is thought to be a crucial building blocks for the polysaccharide GAG production [38], when cultured under glucose-starved condition, GAG^−^ had a fitness advantage in long-term competition but not in short-term competition. Interestingly, this phenotype was also observed under iron-limited environment, although the role of iron on the biosynthesis of GAG is unknown. Therefore, our results indicate that nutrient limitation has a significant effect on the fitness of cheaters in GAG cooperation.

The capacity of GAG^−^ to surpass GAG^+^ in mixed biofilms may indicate that GAG, despite being expensive to generate [39], provides advantages to both the cells that produce it and those nearby. GAG^−^ may take advantage of the GAG created by others while avoiding the energetic expense of its synthesis. Alternatively, it could be a property related to the differential adhesive strengths between GAG^+^ cells and GAG^−^ cells. GAG^+^ cells exhibit higher adhesiveness and have a greater tendency to localize underneath less adhesive cells, where oxygen and nutrients are scarce.

Unlike what one might anticipate regarding a public good, our research showed that GAG^−^ failed to effectively exploit (or cheat) the GAG^+^ under conditions of cell wall stress. These findings align with the competitive advantage hypothesis [7]. When mixed biofilms consist of a substantial amount of GAG^−^, there is a marked increase in susceptibility to caspofungin. Notably, within these biofilms, GAG^+^ demonstrated a greater level of fitness compared to GAG^−^, suggesting that the protective benefits conferred by GAG are primarily advantageous for the cells that produce it. Given that GAG is present both on the surface of *A. fumigatus* hyphae and in the extracellular matrix as a secreted component [26], GAG^−^ cells can easily access the secreted GAG, but lack access to GAG associated with the cell wall. It is possible that there are functional distinctions in caspofungin tolerance linked to the differences between cell wall-associated GAG and secreted GAG. Alternatively, the concentration of GAG may simply be greater in areas that are in close range to GAG^+^, offering a heightened level of protection against caspofungin. Unlike siderophores and the majority of QS-dependent public goods, which are generally diffusible secreted products [36,40], the main role of EPS is to adhere to surfaces and neighboring cells [41]. Previous studies indicated that *P. aeruginosa* PSL primarily localizes around the edges of biofilm microcolonies, encapsulating the cells [42], though some portion is probably diffusible and may function as an intercellular signaling molecule. Thus, GAG likely has limited diffusive properties in the biofilms. Indeed, a significantly greater amount of GAG was detected on the surface of hyphae of GAG^+^ than in that GAG^−^. However, the mechanisms of GAG in the tolerance to cell wall stresses remains unexplored. It is likely that EPS containing GAG has the ability to prevent the penetration of the caspofungin. Another possibility is that, considering GAG is also a component of the cell wall [43], it might play a specific role in maintaining cell wall integrity under cell wall stress.

Biofilms are central to how microbes live and influence humans [12]. The filamentous fungus *A. fumigatus* biofilms present a unique morphology and architecture that set them apart from those associated with bacterial and yeast biofilms. Our study reveals that the social nature of GAG exhibits unique sociological properties, which are different from the previous studies in bacteria. The results of our research, combined with earlier studies, emphasize the essential importance of recognizing the various functions and impacts that different EPS exhibit as social characteristics in biofilms.

## Figures and Tables

**Figure 1 jof-11-00695-f001:**
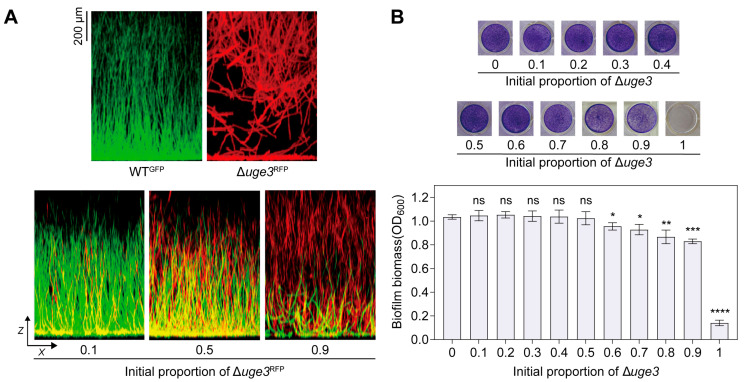
GAG provides social benefits in mixed biofilms. (**A**) Representative confocal images depicting the side view (XZ) of WT^GFP^ biofilms, Δ*uge3*^RFP^ biofilms (upper panels) and their mixed biofilms (lower panels). Conidia of WT^GFP^ and Δ*uge3*^RFP^ were separately cultured to form WT^GFP^ biofilms and Δ*uge3*^RFP^ biofilms. Conidia of WT^GFP^ and Δ*uge3*^RFP^ were co-cultured at indicated ratios to form mixed biofilms. (**B**) Representative crystal violet assay photos (upper panels) and biomass (lower panels) of mixed biofilms at different ratios. Conidia of WT and Δ*uge3* mutant at indicated ratios were statically grown for 24 h in minimal medium (MM). The biofilm biomass was determined by a crystal violet assay. Experiments were conducted a minimum of three times, with each bar indicating the mean ± standard deviation (SD). A one-way analysis of variance (ANOVA) along with multiple comparison tests was utilized for statistical analysis. * *p* < 0.05; ** *p* < 0.01; *** *p* < 0.001; **** *p* < 0.0001; ns represents no significance.

**Figure 2 jof-11-00695-f002:**
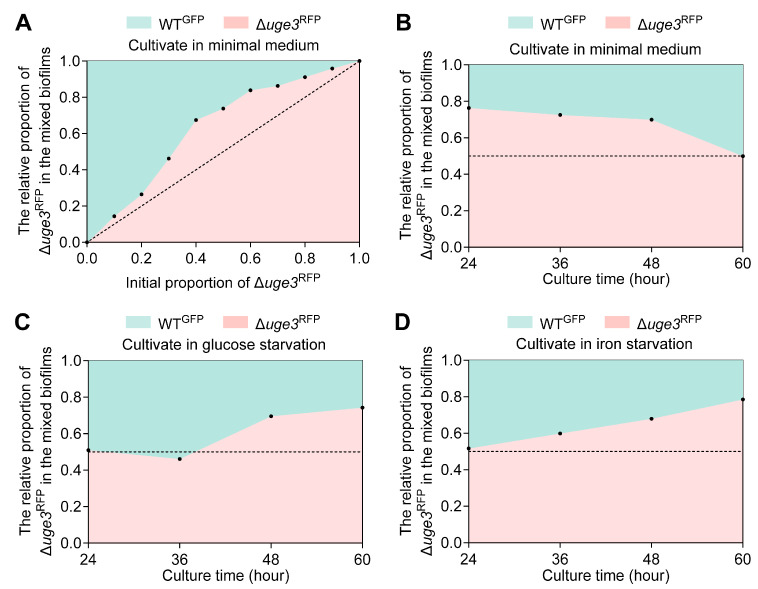
GAG production is an exploitable cooperative behavior in the mixed biofilms. (**A**) The relative fitness of GAG^+^ (WT^GFP^) and GAG^−^ (Δ*uge3*^RFP^) in the mixed biofilms at different ratios. Conidia of WT^GFP^ and Δ*uge3*^RFP^ at indicated ratios were statically grown for 24 h in MM. (**B**) The relative fitness of GAG^+^ and GAG^−^ in the mixed biofilms. Conidia of WT^GFP^ and Δ*uge3*^RFP^ at a 1:1 ratio were statically grown in MM and biofilms were collected at indicated time points. (**C**,**D**) The relative fitness of GAG^+^ and GAG^−^ in the mixed biofilms under nutrient limitation. Conidia of WT^GFP^ and Δ*uge3*^RFP^ at a 1:1 ratio were statically grown in glucose starvation medium (MM containing 0.05% glucose) (**C**) and iron starvation medium (MM without addition of iron) (**D**), biofilms were collected at indicated time points. The relative proportion of Δ*uge3*^RFP^ in the mixed biofilms was quantitated by quantitative PCR.

**Figure 3 jof-11-00695-f003:**
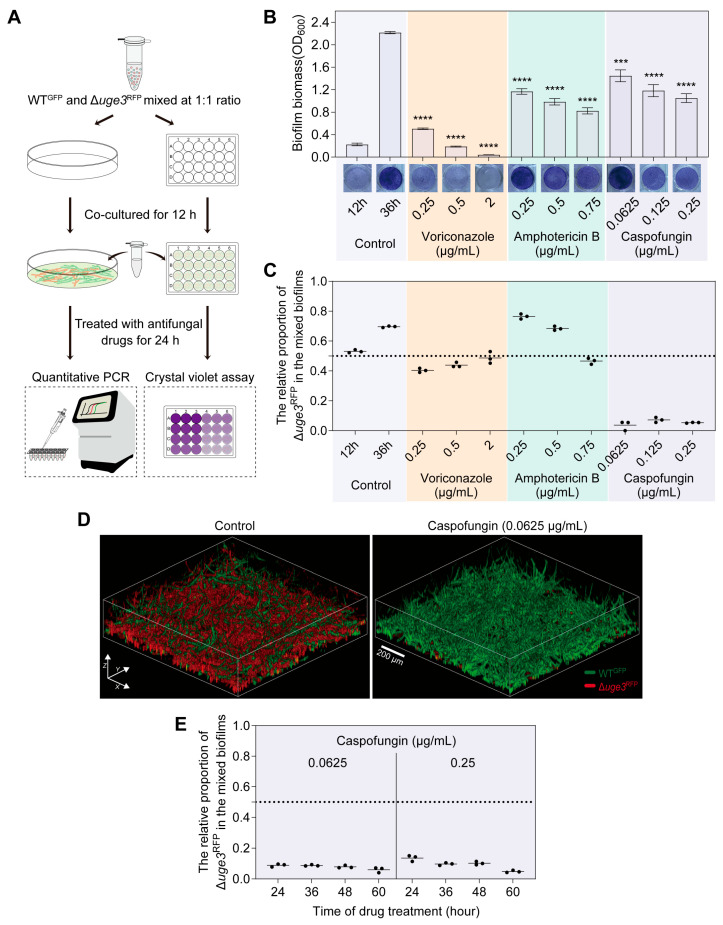
GAG confers a competitive trait under caspofungin treatment in the mixed biofilms. (**A**) Schematic diagram of quantitative PCR and crystal violet assay in mixed biofilms under antifungal drugs treatment. (**B**) Biomass of mixed biofilms under antifungal drug stresses. Conidia of WT and Δ*uge3* mutant at a 1:1 ratio were statically grown for 12 h in MM, then treated with antifungal drugs for an additional 24 h. The biofilm biomass was determined by a crystal violet assay. The untreated 12 and 36 h biofilms were designated as controls. Experiments were conducted a minimum of three times, with each bar indicating the mean ± SD. A one-way ANOVA along with multiple comparison tests was utilized for statistical analysis. *** *p* < 0.001; **** *p* < 0.0001. (**C**) The relative fitness of GAG^+^ and GAG^−^ in the mixed biofilms under antifungal drugs treatment. Conidia of WT^GFP^ and Δ*uge3*^RFP^ at a 1:1 ratio were statically grown for 12 h in MM, then treated with antifungal drugs for an additional 24 h. Untreated 12 and 36 h biofilms were collected as controls. (**D**) Representative confocal images of mixed biofilms under caspofungin treatment. Conidia of WT^GFP^ and Δ*uge3*^RFP^ at a 1:1 ratio were statically grown for 12 h in MM, then treated with caspofungin for an additional 24 h. (**E**) The relative fitness of GAG^+^ and GAG^−^ in the mixed biofilms under caspofungin treatment. Conidia of WT^GFP^ and Δ*uge3*^RFP^ at a 1:1 ratio were statically grown for 12 h in MM, then treated with caspofungin and biofilms were collected at indicated time points. The relative proportion of Δ*uge3*^RFP^ in the mixed biofilms was quantitated by quantitative PCR.

**Figure 4 jof-11-00695-f004:**
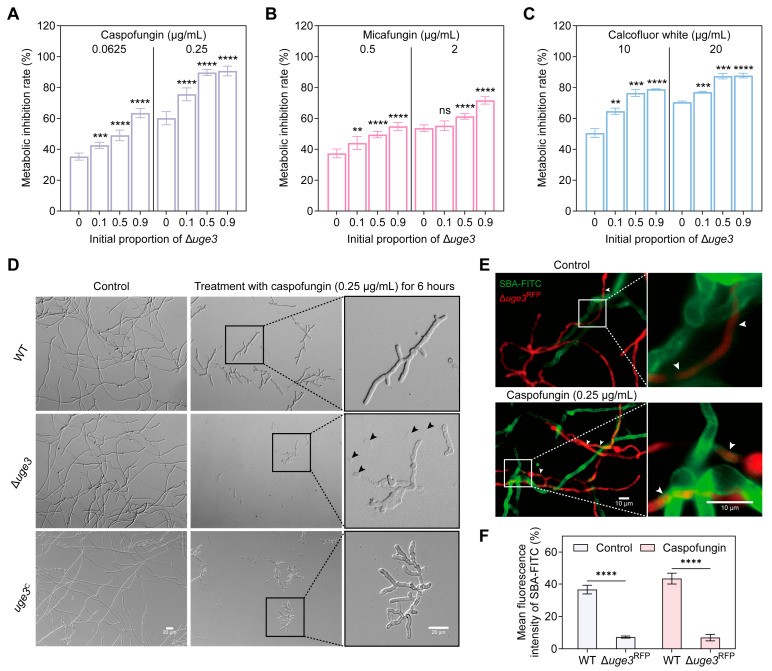
GAG plays protective roles in conferring tolerance to cell wall stresses. (**A**–**C**) The metabolic inhibition assay of mixed biofilms. Conidia of WT and Δ*uge3* mutant at indicated ratios were statically grown for 24 h in MM and then treated with caspofungin (**A**), micafungin (**B**) and calcofluor white (**C**) at the indicated concentrations for an additional 12 h. The metabolic activity after treatments was determined by XTT assay and the percentage of metabolic inhibition compared with non-treated cells. Experiments were conducted a minimum of three times, with each bar indicating the mean ± SD. A one-way ANOVA along with multiple comparison tests was utilized for statistical analysis. ** *p* < 0.01; *** *p* < 0.001; **** *p* < 0.0001; ns represents no significance. (**D**) Representative images of WT, Δ*uge3* mutant and *uge3*^C^ hyphae grown under caspofungin treatment. Conidia of WT, Δ*uge3* mutant and *uge3*^C^ were separately grown for 10 h in MM, then treated with caspofungin for an additional 6 h. The right panels show magnifications of the framed sections in the left panels. Examples of lysed germlings are indicated with black arrowheads. (**E**) Representative images of the indicated strains stained with a GAG-specific fluorescein-tagged soybean agglutinin lectin (SBA-FITC). Conidia of WT and Δ*uge3*^RFP^ at a 1:1 ratio were statically grown for 12 h in MM with and without caspofungin. The right panels show magnifications of the framed sections in the left panels. Arrows indicate GAG on the surface of Δ*uge3*^RFP^ hyphae. (**F**) Quantification of SBA-FITC fluorescence intensity on the surface of WT and Δ*uge3*^RFP^ hyphae at the sites indicated by arrows in (**E**).

**Figure 5 jof-11-00695-f005:**
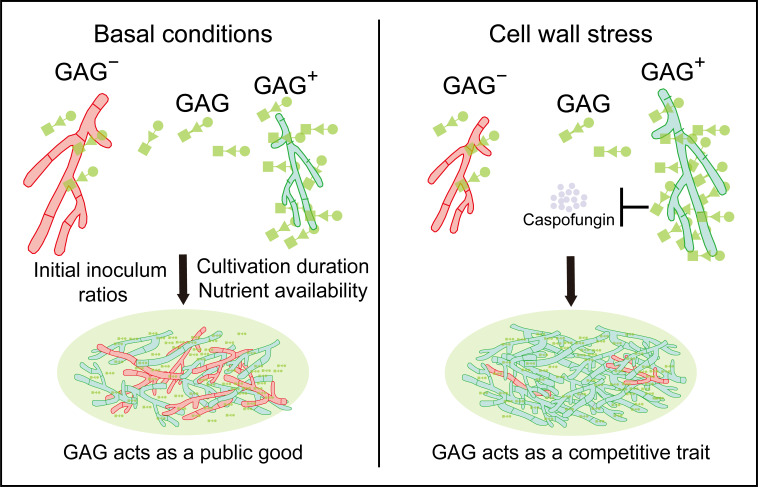
Schematic illustration of GAG displaying context-dependent cooperative and competitive social traits in mixed-biofilms. Under basal conditions, GAG can be shared between GAG^+^ and GAG^−^ in mixed biofilms. This led to significant competitive advantages for GAG^−^, with fitness outcomes dependent on initial inoculum ratios, cultivation duration, and nutrient availability. Under cell wall stress, such as those imposing caspofungin, the majority of GAG retained within GAG^+^ confer them with greater stress tolerance and fitness advantage compared to GAG^−^.

## Data Availability

The original contributions presented in this study are included in the article/Appendix A. Further inquiries can be directed to the corresponding author.

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
