# Peer review of "The Aspergillus fumigatus Extracellular Polysaccharide Galactosaminogalactan Displays Context-Dependent Cooperative and Competitive Social Traits in Mixed Biofilms"

_jof, 2025, doi:10.3390/jof11100695_

Round 1

Reviewer 1 Report

The manuscript "The Aspergillus fumigatus extracellular polysaccharide galactosaminogalactan displays context-dependent cooperative and competitive social traits in mixed biofilms" submitted for review is devoted to the analysis of the features of the participation of galactosaminogalactan in the realization of social relationships in the biofilm of Aspergillus fumigatus. The text makes a very favorable impression, it is clearly structured and written, the illustrative material is of excellent quality, the results obtained by the authors are quite interesting, the conclusions are justified and placed in the context of the findings of other authors. A manuscript is one of the rare examples where there is a desire to recommend a text for publication with practically no changes.

After carefully reviewing the manuscript, I came to the conclusion that it can be published in its present form.

Reviewer 2 Report

The manuscript is well written and the results are clearly presented.   There are a few minor changes that I recommend in the detailed comments below.

There are several small changes that I would recommend to enhance the presentation of the data in the manuscript.

1) I would recommend adding a Figure showing the construction of the different gene constructs used for creating the strains used in the analysis.  I would think it could be added to the supplementary materials and would help the reader navigate section 2.2 where all of the primers and PCR reactions are given.

2) Page 2 line 51.  Some readers might not know what PSL is and the full name of the polysaccharide ought to be included when PSL is first mentioned.

3) Page 3 line 91.   A reference for the composition of minimal medium is needed.

4) Page 3 line 98.   A mention that the uge3 encodes the GAG synthase should be included in this sentence.

5) Page 7 Figures 2.  Having the Figure filled in with green and red color gives the impression that the relative proportion of uge3 deletion cells is being measure across the continuum of values (initial concentrations or culture time).   I would recommend that a circle or large dot be placed in the Figures for the values actually measured (initial concentrations of 0.0, 0.2, 0.4, 0.6, 0.8 and 1.0 in Figure 2A and 24hr, 36hr, 48hr, and 60hr in Figures 2B, 2C, and 2D).

6) Page 9 lines 337-348.  Reading lines 348 -352 indicates that uge3 deletion mutant and wildtype cells are equally susceptible the caspofungin when grown of agar plates.  Lines 337-348 indicate that the uge3 deletion mutant is much more susceptible to caspofungin and reading the figures it seems the assay in which the mutant is more susceptible is for conditions for biofilm formation.  I recommend that the authors make some changes to the text lines 337-348 to emphasize that the susceptibility is in a medium for biofilm formation.

7) Page 10 line 369.  The figure legend for Figure 4 reads "grown under treated with caspofungin" and the sentence needs to be rewritten.

8) Page 10 Figure 4E.  When I look at Figure 4E I can see the GAG present on uge3 deletion mutant cells as indicated by the arrowheads.  However, when I look at the Figures, it looks to me like significant regions of the uge3 deletion mutant cells do not have any associated GAG.  I would recommend that the authors note this fact in their discussion of Figure 4E.  Perhaps the authors have looked at enough images at high magnification to convince them that most of the deletion mutant cell walls are decorated with GAG.  In either case, as brief sentence indicating what proportion of the mutant cell walls have associated GAG should be added to the text.

9) Pag11 line 389.   The sentence "A working model showing the social nature of GAG in the mixed biofilms was depicted (Figure 5)." should be changed to read "is depicted in Figure 5".  

Reviewer 3 Report

Biofilms formed by bacteria and fungi provide them with competitive advantages when colonizing certain biotopes and when exposed to adverse environmental factors. However, the ability to form biofilms and the associated resistance in organisms pathogenic to humans and animals pose a complex challenge for the treatment of the infections they cause. The authors of this article examine the possible social role of galactosaminogalactan, a key component of the biofilm matrix of the pathogenic fungus Aspergillus fumigatus. The biosynthesis of this extracellular polymer has been extensively studied, and its role in the formation of Aspergillus fumigatus biofilms has been demonstrated, confirming the relevance of studying its role in mixed biofilms with bacteria. The authors chose a mutant strain incapable of EPS production and its complement as their study subjects. This approach allowed them to evaluate the social nature of the polymer, highlighting the scientific novelty of this study. To address these issues, they successfully use differential labeling of mutant and wild-type fungi with fluorescent proteins, in parallel with the classical approach of visualizing biofilm formation using crystal violet. I must admit that I read the manuscript with great interest and attention. The work is well illustrated. I would especially like to highlight the confocal images, which clearly confirm the obtained results. The experimental block using fungicides of different structures raises some questions for me. I may disagree with the authors regarding some of their conclusions about the social nature of EPS in biofilms, but I understand the logic of their reasoning. I am confident that this work will generate interest among readers and will be actively discussed.

I have a few minor comments and questions for the authors.

Introduction. I believe the authors should focus on the specific features of biofilm formation and functioning, including mixed ones, of Aspergillus fumigatus.

Section 3.2 and Fig. 2. The observed advantage of the mutant strain under growth conditions deficient in certain components may be related to the greater requirement of the wild-type fungus for EPS formation. However, the slight predominance of the mutant strain can be assessed as a competitive advantage during a specific period of time, for example, 60 hours of cultivation. However, to more accurately assess the competitive advantage, a longer cultivation period would be necessary. Will the mutant strain's advantage persist if the biofilms are transferred to conditions favorable for growth?

Section 3.3, lines 294–298. It seems somewhat inaccurate to me that the authors' explanation is that caspofungin treatment cannot completely eliminate GAG- from biofilms. GAG- remain in the biofilm, since GAG+ secreted by the EPS can, as has been shown, play a protective role. It is entirely logical that, during biofilm matrix formation, hyphae of the mutant strain are drawn into it, and some, albeit insignificant, percentage of them survive. I believe the wording in this section and in the discussion of these results should be somewhat softened.
